# Urothelial Oxidative Stress and ERK Activation Mediate HMGB1-Induced Bladder Pain

**DOI:** 10.3390/cells12101440

**Published:** 2023-05-22

**Authors:** Shaojing Ye, Dlovan F. D. Mahmood, Fei Ma, Lin Leng, Richard Bucala, Pedro L. Vera

**Affiliations:** 1Lexington VA Health Care System, Research & Development, Lexington, KY 40502, USA; 2Department of Internal Medicine, Yale University, New Haven, CT 06510, USA; 3Department of Physiology, University of Kentucky, Lexington, KY 40506, USA

**Keywords:** bladder pain, MIF KO, HMGB1, ROS, ERK

## Abstract

Activation of intravesical protease activated receptors-4 (PAR4) results in bladder pain through the release of urothelial macrophage migration inhibitory factor (MIF) and high mobility group box-1 (HMGB1). We aimed to identify HMGB1 downstream signaling events at the bladder that mediate HMGB1-induced bladder pain in MIF-deficient mice to exclude any MIF-related effects. We studied whether oxidative stress and ERK activation are involved by examining bladder tissue in mice treated with intravesical disulfide HMGB1 for 1 h and analyzed with Western blot and immunohistochemistry. HMGB1 intravesical treatment increased urothelium 4HNE and phospho-ERK1/2 staining, suggesting that HMGB1 increased urothelial oxidative stress and ERK activation. Furthermore, we examined the functional roles of these events. We evaluated lower abdominal mechanical thresholds (an index of bladder pain) before and 24 h after intravesical PAR4 or disulfide HMGB1. Intravesical pre-treatments (10 min prior) included: N-acetylcysteine amide (NACA, reactive oxygen species scavenger) and FR180204 (FR, selective ERK1/2 inhibitor). Awake micturition parameters (voided volume; frequency) were assessed at 24 h after treatment. Bladders were collected for histology at the end of the experiment. Pre-treatment with NACA or FR significantly prevented HMGB1-induced bladder pain. No significant effects were noted on micturition volume, frequency, inflammation, or edema. Thus, HMGB1 activates downstream urothelial oxidative stress production and ERK1/2 activation to mediate bladder pain. Further dissection of HMGB1 downstream signaling pathway may lead to novel potential therapeutic strategies to treat bladder pain.

## 1. Introduction

We recently reported that the activation of intravesical PAR4 receptors leads to the release of urothelial macrophage migration inhibitory factor (MIF) and high mobility group box-1 (HMGB1) in vivo and human urothelial cells in vitro [1]. MIF and HMGB1 are considered alarmins or atypical chemokines that perform a variety of extracellular functions once released into the extracellular space [2]. HMGB1 is located in urothelial cells [3] and is a nuclear protein released (actively or passively) into the extracellular space and can exist in multiple redox forms to mediate inflammation and pain [4,5].

HMGB1 has also been demonstrated to mediate bladder pain in other rodent models of bladder pain (including inflammatory models), since the blockade of HMGB1 or HMGB1 receptors inhibited bladder pain [6,7,8,9]. In addition, we recently showed that levels of released HMGB1 are correlated with PAR4-induced bladder pain [10] and HMGB1 plays a pivotal role in mediating pain in this model since systemic treatment with glycyrrhizin (HMGB1 antagonist) prevents acute PAR4-induced bladder pain [3] while systemic treatment with ethyl pyruvate (to prevent HMGB1 release) decreased established PAR4-induced bladder pain [10].

Since antagonism of MIF receptors blocks HMGB1 release in vivo and in vitro [1], HMGB1 is downstream of MIF in PAR4-mediated bladder pain (Figure 1). Intravesical disulfide HMGB1 (but not the thiol form) induced bladder pain in wild-type mice and MIF-deficient mice [11] through the activation of TLR4 receptors [12]. Figure 1 is a summary of our current understanding of the events involved in PAR4-induced bladder pain. TLR4 receptors activate multiple signaling events including reactive oxygen species (ROS) formation and extracellular signal regulated kinases (ERK) phosphorylation as reported in different systems [13,14,15,16,17,18]; however, their role in HMGB1-mediated bladder pain is not known.

We recently showed [10] that, in WT mice, PAR4 elicited bladder pain that was mediated by HMGB1 and oxidative stress. Investigators of other systems have shown that both MIF and HMGB1 can activate ERK pathways (MIF: [19,20]; HMGB1: [21,22]) and/or oxidative stress (MIF: [23,24]; HMGB1: [22,25,26]. Therefore, since our previous results in WT mice regarding oxidative stress could be explained by either MIF or HMGB1, the present study used MIF deficient mice to isolate the effects observed strictly to HMGB1 and not to possible MIF-related effects. We aimed to examine whether (1) intravesical HMGB1 activated bladder oxidative stress and ERK phosphorylation and (2) antagonism of these changes modulated HMGB1-induced bladder pain measured one day after treatment.

## 2. Materials and Methods

All animal experiments were approved by the Lexington VA Health Care System Institutional Animal Care and Use Committee (VER-19-005-AF) and performed according to the guidelines of the National Institutes of Health. MIF knockout (KO) mice were obtained from Yale University and genotyped as described [27]. Briefly, DNA was isolated from tail tip biopsies and PCR was performed to determine MIF deletion by using the following primers: MIF deletion forward, 5’-GAA TGA ACT GCA GGA CGA GG-3’. MIF deletion reverse 5’-GCT CTT CGT CCA GAT CAT CC-3’; WT forward, 5’-ACG ACA TGA ACG CTG CCA AC-3’; WT forward, 5’-ACC GTG GTC TCT TAT AAA CC-3’. PCR products were 300 bp for the MIF deletion allele and 200 bp for the WT allele. Homozygous mice were bred. Female mice at 12–14 weeks old were used for all experiments. Mice were housed in a normal light cycle (14 h on; 7 am; 10 h off; 9 pm) with free access to food (Teklad Global 2018, Envigo, Indianapolis, IN, USA) and water.

### 2.1. Experiment 1: Oxidative Stress and ERK Phosphorylation Changes in Our Bladder Pain Model

We used disulfide HMGB1 to stimulate acute bladder hyperalgesia (BHA) as described earlier [11]. We tested whether intravesical HMGB1 affected reactive oxygen species production and ERK phosphorylation in the bladder by measuring levels of 4-hydroxynonenal (4HNE; a widely used marker of oxidative stress [28]) and levels of phosphorylated ERK, respectively, using Western blotting and immunohistochemistry.

Briefly, we anesthetized female mice using isoflurane (3% induction, 1.5% maintenance) and inserted a transurethral catheter (PE10, 11 mm length). We drained urine from the bladder by applying gentle pressure to the lower abdomen. We performed two sets of experiments using this model.

Bladders of two groups of mice (N = 5/group) were slowly instilled with 50 μL of either PBS (vehicle control) or 10 μg HMGB1 (to induce bladder pain) and remained in the bladder for 1 h. After the instillation, bladders were removed and sectioned in half longitudinally. One half was frozen (−80${}^{\circ }$C) for Western blotting (described below) while the detrusor body (excluding the bladder base) of the other half was placed in 4% buffered formaldehyde and embedded in paraffin for hematoxylin and eosin (H&E) staining and immunohistochemistry (described below).

#### 2.1.1. Western Blotting

Bladder tissue from intravesical disulfide HMGB1 and control (PBS) treated-mice was homogenized in RIPA Buffer (BP-115, Boston BioProducts, Milford, MA, USA) with 1x protease and phosphatase inhibitors. The total protein concentration was determined by BCA assay. Equal amounts of protein (20 μg) were separated using a 4–15% Mini-PROTEAN TGX precast polyacrylamide gel (Bio-Rad, Hercules, CA, USA) and transferred to a polyvinylidene difluoride membrane. After blocking with 3% BSA in tris-buffered saline (TBS) for one hour, the membranes were exposed to primary antibodies (anti-4HNE rabbit polyclonal antibody; cat. no. 393207-M from Sigma-Aldrich (St. Louis, MO, USA) at 1:2000 dilution, anti-phospho-ERK1/2 rabbit monoclonal antibody, cat. no. 4370 from cell signaling at 1:2000 dilution, anti-ERK1/2 rabbit monoclonal antibody; cat. no. 4695 from cell signaling, 1:1000 dilution) at 4 °C overnight with shaking, followed by incubation with a biotinylated anti-rabbit secondary antibody (BA-1000, Vector Labs, Newark, CA, USA) and streptavidin-HRP conjugates (Cat. no. 21140, Thermo Scientific, Waltham, MA, USA), along with GAPDH as a loading control (A00915, GenScript at 1:2500 dilution). Immunoreactive bands were visualized using an enhanced chemiluminescence kit (SuperSignal West Femto; Thermo Scientific) and band densitometry was performed using an image analysis tool (BioRad, Hercules, MA, USA).

#### 2.1.2. Immunohistochemistry

Paraffin sections (5 μm) of paraformaldehyde fixed bladders were dewaxed and processed for antigen retrieval using citrate buffer, pH 6.0 at 94 °C for 20 min. Sections were treated with 3% H_2_O_2_ for 10 min to quench endogenous peroxidase and then with normal donkey serum (3% in TBS) for 1 h and then incubated with primary antibody (rabbit anti-4HNE at 1:400 or rabbit anti-phospho ERK1/2 at 1:2000 overnight at 4 ${}^{\circ }$C). Bound primary antibody was detected using a VECTASTAIN^®^ ABC Kits (PK-4001, Vector Labs). Digital images were captured with a Leica (Wetzlar, Germany) microscope.

#### 2.1.3. Histological Measurements

Bladder paraffin sections (5 μm) were processed for routine hematoxylin and eosin (H&E) staining. The stained sections were evaluated by two independent observers blinded to the experimental groups’ assignment and scored separately for edema and inflammation according to the following scale: 0, no edema/no infiltrating cells; 1, mild submucosal edema/few inflammatory cells; 2, moderate edema/moderate number of inflammatory cells; 3, frank edema, vascular congestion/many inflammatory cells, as per our previous studies [1,3,10,11,12,29].

### 2.2. Experiment 2: Oxidative Stress and ERK Phosphorylation Blockage in HMGB1-Induced Bladder Hyperalgesia

In this experiment, we studied the effect of intravesical infusion of different antagonists (pre-treatment) on HMGB1-provoked BHA. Isoflurane was used to anesthetize mice as described above and the bladders of separate groups of mice were slowly instilled with 50 μL of each pre-treatment (Figure 2). The following substances were administered 10 min prior to the instillation of PAR4-activating peptide (PAR4-AP; AYPGKF-NH2), scrambled peptide (as negative control) or 10 μg HMGB1: (1) Solvents (PBS; Methyl cellulose, Methcell); (2) N-Acetylcysteine amide (NACA, 3 mg, in PBS) as an ROS scavenger [30,31]; (3) FR180204 (200 μg, in 0.1% Methcell/PBS), a selective ERK1/2 phosphorylation inhibitor [32]. The peptides (10 μg for HMGB1 or 100 μM for PAR4-AP/scrambled peptide) were dissolved in sterile PBS (pH 7.4, 100 μL) and remained in the bladder for 1 h. After the installation, mice were allowed to recover from anesthesia and returned to their cages.

#### 2.2.1. Abdominal Mechanical Sensitivity

Mice were acclimated to the testing conditions as follows:(Days 1, 2) Acclimation to testing room:– Mice placed in testing room and left undisturbed for 3 h;(Days 3, 4) Acclimation to testing chamber:– Mice placed in testing chamber and left undisturbed for 2 h;(Day 5) Acclimation to von Frey (VF) monofilaments:– Mice placed in testing chamber and VF monofilaments applied to lower abdominal area;(Day 7) Baseline VF testing.

Testing of lower abdominal mechanical hypersensitivity (an index of bladder pain) was performed as previously described [1,10,33,34]. Briefly, a 50% mechanical threshold [35] was calculated by measuring the response to VF fibers (0.008, 0.02, 0.07, 0.16, 0.4, 1.0, 2.0 and 6.0 g) applied to the lower abdominal region. A positive response was defined as any one of three behaviors: (1) licking the abdomen, (2) flinching/jumping, or (3) abdomen withdrawal. Whenever a positive response to a stimulus occurred, the next smaller von Frey filament was applied. Otherwise, the next higher filament was applied. The 50% thresholds were measured at baseline (prior to any treatment) and approximately 24 h after bladder pre-treatments and treatments.

#### 2.2.2. Micturition Parameters in Awake Mice

We measured micturition volume and frequency in awake mice using the Voided Stain on Paper (VSOP) method [36] as described earlier [1,10]. Briefly, 24 h after intravesical treatment and after testing the abdominal mechanical threshold, mice were placed in a plastic enclosure individually with freedom to move around and access to water. Filter paper was placed under the cage to collect urine during a 3-h observation period. Micturition volumes were determined by linear regression using a set of known volumes. Micturition frequency is reported as the number of micturitions per 3-h observation period.

#### 2.2.3. Histological Measurements

As a final step in the study, we reanesthetized the mice with 3–4% isoflurane and after a lower abdominal incision, we removed the bladders. Bladder segments were fixed with 4% buffered paraformaldehyde and after paraffin embedding, 5 μm sections were stained with H&E and scored for edema and inflammation as described above.

### 2.3. Reagents

PAR4-AP (AYPGKF-NH2) and corresponding scrambled peptide (YAPGKF-NH2) were from Peptides International, Inc. (Louisville, KY, USA). Disulfide HMGB1 peptides were from HMGBiontech (Milano, Italy). N-Acetylcysteine amide was purchased from Tocris (Minneapolis, MN, USA). FR180204 was from Sigma-Aldrich. H&E staining reagents were from Fisher Scientific. The rest of the materials used were from Sigma-Aldrich or as described in the methods.

### 2.4. Statistical Analysis

All statistical analyses were performed using R [37].

We evaluated differences in 50% von Frey threshold scores using a repeated measures two-way analysis of variance (ANOVA) with time and treatment as factors (rstatix package). We further evaluated simple main effects at each time point using ANOVA followed by Dunnett’s test once the ANOVA reached significance. A *p* < 0.05 was considered statistically significant. Mean and ±SEM are reported.

## 3. Results

### 3.1. Experiment 1: Changes in Bladder Oxidative Stress and ERK Phosphorylation by Intravesical HMGB1 in MIF KO Mice

We examined changes in bladder levels of 4HNE as a marker for oxidative stress and ERK phosphorylation using both Western blotting and immunohistochemistry (IHC).

Treatment with intravesical HMGB1 for 1 h resulted in an increase in levels of 4HNE in the bladder, as measured by Western blotting, when compared to intravesical treatment with PBS (Figure 3A). Densitometry analysis showed this increase to be statistically significant (Figure 3B). Furthermore, treatment with HMGB1 increased 4HNE immunostaining over the entire bladder and particularly over the urothelium when compared to PBS treatment (Figure 3C,D). Thus, our results document an increase in 4HNE levels in the bladder after intravesical HMGB1 that is mostly localized to the urothelium.

Similarly, treatment with HMGB1 resulted in an increase in pERK1/2 levels in the bladder when compared to PBS treatment, while levels of total ERK were not affected by intravesical treatments (Figure 4A). Densitometry analysis and comparison of the ratio of pERK1/2 to total ERK showed there was a statistically significant elevation after HMGB1 treatment (Figure 4B). We also examined pERK1/2 immunostaining after PBS and HMGB1 intravesical treatments. After PBS treatment, pERK1/2 immunostaining was mostly nuclear and restricted to the basal and intermediate layers of the urothelium (Figure 4C). Intravesical treatment with HMGB1, on the other hand, resulted in a pronounced increase in pERK1/2 immunostaining in the urothelium and a shift to cytoplasmic localization (Figure 4D).

We examined the effects of one-hour intravesical treatments on bladder histology by evaluating edema and inflammation in H&E bladder sections. None of the bladders treated with intravesical PBS showed signs of edema or inflammation (edema = 0, inflammation = 0; Figure 5, left panel). On the other hand, three (of the six) bladders treated with intravesical HMGB1 showed mild edema and inflammation (edema = 1.4 ± 0.6, inflammation = 1.1 ± 0.5; Figure 5, right panel). These group differences did not reach the level of statistical significance when compared to PBS treatment (*t*-test p=0.07 for both edema and inflammation).

The results from these experiments indicate that intravesical treatment with HMGB1 for 1 h results in increased oxidative stress and ERK phosphorylation involving the urothelium, with histological changes suggesting mild edema and inflammation.

### 3.2. Experiment 2: Bladder Oxidative Stress and ERK Phosphorylation Antagonism on HMGB1-Induced BHA

We then tested the effects of intravesical antagonists to the formation of reactive oxygen species or ERK activation on HMGB1-induced bladder hyperalgesia (BHA). Figure 6 shows the results of testing for changes in the lower abdominal mechanical threshold between baseline (Figure 6A) and 24 h post intravesical administration (Figure 6B).

First, we tested the effects of the intravesical administration of a scrambled peptide or PAR4-AP in MIF KO mice. Compared to baseline, neither scrambled peptide nor PAR4 resulted in a change in the 50% VF threshold 24 h after administration (Figure 6B). However, intravesical HMGB1 dramatically reduced the 50% VF threshold 24 h after treatment and this is interpreted as an index of BHA in both WT and MIF KO mice. These findings agree with our earlier observation that PAR4 does not cause BHA in MIF KO mice, but HMGB1 does in WT and MIF KO mice [11,12].

We studied the efficacy of intravesical pre-treatments with antagonists of ROS formation or ERK1/2 activation by using HMGB1-induced BHA in MIF KO mice. No significant differences were observed in the VF threshold at baseline across all treatment groups (Figure 6A). At 24 h, however, significant treatment effects were noted on the VF threshold (Figure 6B). Repeated measures ANOVA showed a significant effect of time (baseline vs. 24 h; F = 1151, *p* = 2${}^{-16}$ and treatment (F = 19.5, *p* = 4.6${}^{-11}$).

Intravesical pre-treatment with an ROS inhibitor (NACA) or with an ERK1/2 inhibitor (FR180204) partially prevented HMGB1-induced BHA (Figure 6B) while there was no effect on preventing HMGB1-induced BHA in mice pre-treated with intravesical PBS or Methcell (used as solvents).

#### 3.2.1. Effect of Treatments on Awake Micturition Parameters

We measured awake micturition volume for all treatment groups using VSOP. An ANOVA comparing micturition volume across all treatment groups was not statistically significant (F = 1.8, *p* = 0.1); therefore, there were no differences in micturition volume due to treatment (Table 1).

Similarly, we measured awake micturition frequency in all treatment groups. ANOVA was not significant (F =1.02, *p* = 0.4) and therefore there were no treatment effects on micturition frequency (Table 1).

#### 3.2.2. Effect of Treatment on Histological Evidence of Bladder Inflammation

Bladder inflammation was assessed using H&E sections and was scored blinded to treatment for evidence of edema and inflammation. Figure 7 shows representative bladder sections from each group. ANOVA for edema across all treatment groups was not significant (F = 2.3, *p* = 0.05). Post-hoc tests comparing all treatment groups to the “no pain” control group (PBS pre-treatment scrambled peptide treatment) did not show any significant differences (Table 2). There were no significant differences in inflammation across all treatment groups (F = 1.98, *p* = 0.08).

## 4. Discussion

There is accumulating evidence using different animal models showing that HMGB1 is involved in mediating acute and persistent pain both at the periphery and spinal cord levels [4,5]. In fact, HMGB1 was reported to mediate bladder pain in inflammatory models previously [6,9]. HMGB1 is also a crucial component in the sequence of events triggered by the activation of urothelial PAR4 receptors, resulting in bladder pain [1,3,10,12]. In fact, we previously reported that HMGB1 induced bladder pain through the activation of TLR4 receptors [12].

PAR4 induces the release of urothelial MIF and HMGB1, and both of these substances are associated with downstream signaling events (including oxidative stress and ERK activation [2,38,39]). MIF-deficient mice did not show bladder pain after intravesical PAR4 but did show bladder pain after intravesical HMGB1 [11]. Therefore, we aimed to explore the mechanisms of HMGB1-induced bladder pain and restricted our observations only to HMGB1-mediated effects by using MIF-deficient mice. Since TLR4 signaling results in oxidative stress and ERK phosphorylation [40], we examined the role of both of these events in HMGB1-mediated bladder pain in MIF-deficient mice.

First, we observed that, one hour after intravesical treatment with HMGB1, both oxidative stress (as measured by 4HNE levels) and ERK activation (as measured by the ratio of pERK/total ERK) increased in the bladder and particularly over the urothelium when compared to the control treatment (PBS).

Second, we observed that intravesical pre-treatment with an ROS scavenger (NACA) or an antagonist to ERK activation (FR180204) was capable of partially preventing HMGB1-induced bladder pain in MIF KO animals. Our findings are in agreement with Hiramoto et al. [9], who reported that pre-treatment with N-acetylcysteine partially inhibited cyclophosphamide-induced BHA. It should also be noted that we also confirmed our earlier findings that PAR4 did not produce BHA in MIF-deficient mice [11], confirming that HMGB1 is downstream of MIF in this bladder pain model.

We previously reported that the activation of intravesical PAR4 receptors in wild-type mice (or human epithelial cells) leads to the release of urothelial MIF and HMGB1 to induce bladder pain [1,3,29] (Figure 8A). PAR4-induced bladder pain was also mediated by oxidative stress [10]. Since both MIF and HMGB1 (downstream of urothelial PAR4 activation) are involved in mediating ERK activation and oxidative stress [19,20,21,22,23,24,25,26,26], it was not possible to accurately identify our previous results as being MIF- or HMGB1-mediated. Thus, our current study using MIF-deficient mice clearly ruled out a contribution from MIF to our results. This is an important point that further defines our understanding of the signaling mechanisms in the PAR4/MIF/HMGB1/bladder pain pathway by clearly ascertaining only HMGB1-mediated events.

Therefore, our results clearly indicate that intravesical HMGB1 (presumably through TLR4 receptors) triggers bladder oxidative stress and an increase in ERK phosphorylation, and these events are part of the process that results in bladder pain measured one day after HMGB1 stimulation (Figure 8B). Other investigators have shown that inflammatory stimuli or bladder overinflation results in urothelial oxidative stress and/or ERK activation [41,42,43,44]. Our findings suggest that urothelial oxidative stress and ERK activation are implicated in bladder pain in this model. Interestingly, effective intravesical treatment in patients with Interstitial cystitis/bladder pain syndrome (IC/BPS) was correlated with lower levels of a urinary marker for oxidative stress [45].

IC/BPS is a painful condition of unknown etiology affecting over 1 million people and often resulting in poor quality of life [46]. A hallmark symptom of this syndrome is pain often referred to the bladder [46,47]. Two main phenotypes are recognized clinically: patients with ulcerative lesions (Hunner lesions) and inflammation in the bladder and those without [48,49,50]. These two subtypes are not different with regards to pain symptoms [47].

Protease-activated receptors (PAR) 1 through 4 are located in the urothelium [51]. The activation of intravesical PAR4 receptors is a rodent model of bladder pain with minor or no inflammatory changes in the bladder and no changes in micturition parameters [1,3,29,52]. Therefore, this rodent model of bladder pain may be comparable to non-Hunner type IC/BPS, where patients do not display histological signs of bladder inflammation.

Altogether, our findings confirm the important role of MIF and downstream activation of HMGB1 in the bladder to result in bladder pain. It should be pointed out that, similar to intravesical PAR4 [1,3,29,52], intravesical HMGB1 also results in bladder pain with no (or little) histological evidence of bladder inflammation as seen in the current study and our earlier study [11]. In fact, mild edema and inflammation was observed in half of the HMGB1-treated bladders while this was not the case for PBS-treated bladders one hour after treatment and these changes resolved after 24 h while pain still persisted in HMGB1-treated mice.

Tanaka et al. [6] reported that blocking HMGB1 prevented bladder pain but not bladder inflammation in their cyclophosphamide-treated rodents. A different model (intravesical substance P) was also reported to induce bladder pain without major bladder damage and bladder pain was blocked by antagonizing HMGB1 or HMGB1 receptors [8]. Our current results also show that HMGB1 directly causes bladder pain with little or no bladder inflammation. Therefore, there is sufficient evidence to suggest that it is possible to separate inflammatory effects from pain effects at the bladder. In fact, a recent report showed that TLR4 mediates pain not inflammation in an auto-immune bladder inflammation model [53].

In our intravesical PAR4 model of bladder pain, released MIF binds urothelial MIF receptors (e.g., CD74 or CXCR4 [54]) to elicit the release of urothelial HMGB1 [10]. Released HMGB1 binds to intravesical TLR4 receptors [12] to result in the activation of ERK1/2 and reactive oxygen species leading to bladder pain in the absence of bladder inflammation (as determined by histological analysis or inflammatory cytokine expression) [10]. The current studies extend our understanding of the signaling mechanisms involved in this PAR4-MIF-CD74/CXCR4-HMGB1-TLR4 model by providing evidence that oxidative stress and ERK activation downstream of HMGB1 play a part in mediating bladder pain (Figure 8C). Since our treatments to block oxidative stress or ERK activation partly (albeit significantly) prevented HMGB1-induced bladder pain, it is reasonable to propose that other TLR4-signaling events may also be involved and this warrants further investigation.

The fact that there is little or no inflammation in our PAR4-(and HMGB1-)mediated bladder pain model allows us to focus exclusively on pain mechanisms at the bladder. Our rodent model may mimic non-ulcerative IC/BPS where no Hunner lesions or bladder inflammation are documented and which represent most of the IC/BPS patients [47,55]. The elucidation of the mechanisms of bladder pain separated from bladder damage may prove useful in increasing our understanding of the disease process in non-ulcerative IC/BPS patients and may lead to the discovery of novel therapeutic targets to alleviate bladder pain in these patients.

## 5. Conclusions

In summary, our study showed that intravesical HMGB1 independently stimulated bladder pain in MIF null mice. Urothelial ERK1/2 was activated while urothelial oxidative stress was induced post HMGB1 instillation. Systemic inhibition of ERK and ROS significantly reduced bladder pain in mice. Further dissecting HMGB1 signaling pathways involving ERK and ROS may develop novel therapeutic targets for treating bladder pain conditions such as IC/BPS.

## Figures and Tables

**Figure 1 cells-12-01440-f001:**
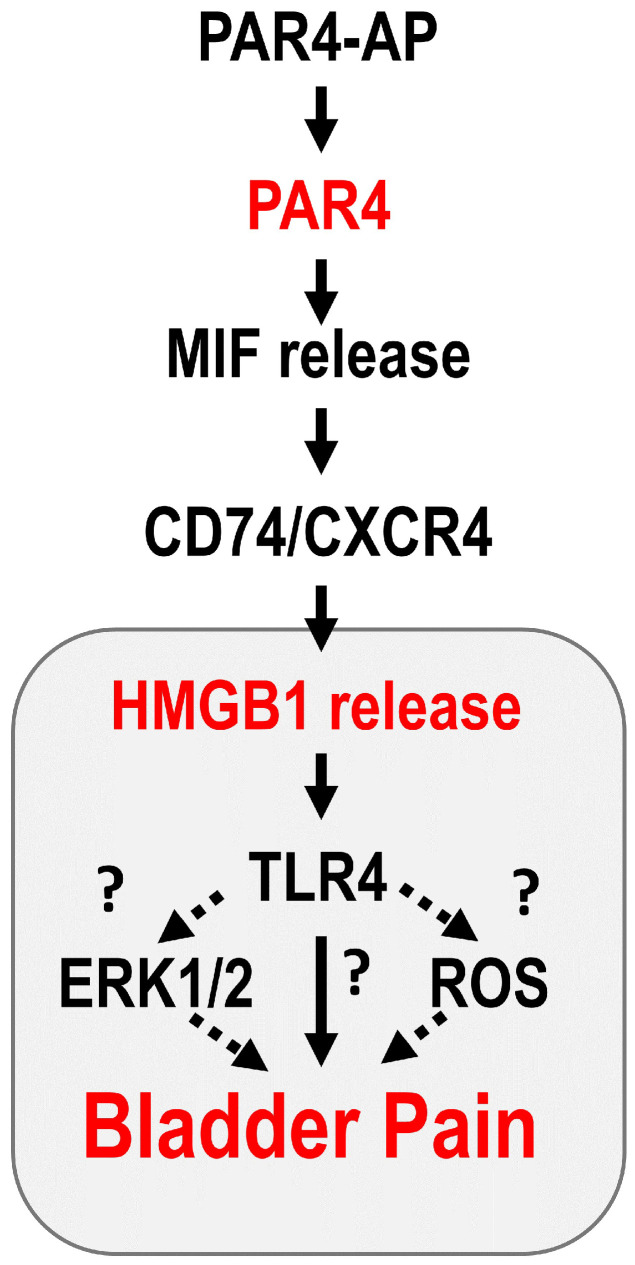
HMGB1 signaling pathway in PAR4-induced bladder pain.

**Figure 2 cells-12-01440-f002:**
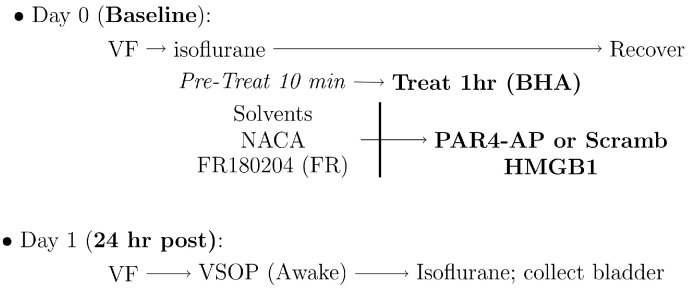
Diagram of experimental scheme for studies in Experiment 2. On day 0, a baseline abdominal von Frey (VF) measurement was performed before MIF KO mice were anesthetized with isoflurane. Bladder was emptied through a transurethral catheter and then solvents—NACA or FR180204—were instilled for 10 min prior to instillation of PAR4 activating peptide (PAR4-AP), PAR4 scrambled peptide (Scramb), or disulfide HMGB1 (to elicit BHA) for an hour. After that, the bladders were emptied and mice were allowed to recover. After 24 h (on Day 1), VF measurement was tested, and then micturition volume and frequency were recorded in awake mice using the Voided Stain on Paper Method (VSOP) for 3 h. At the end of the study, bladders were collected for histology and further analysis.

**Figure 3 cells-12-01440-f003:**
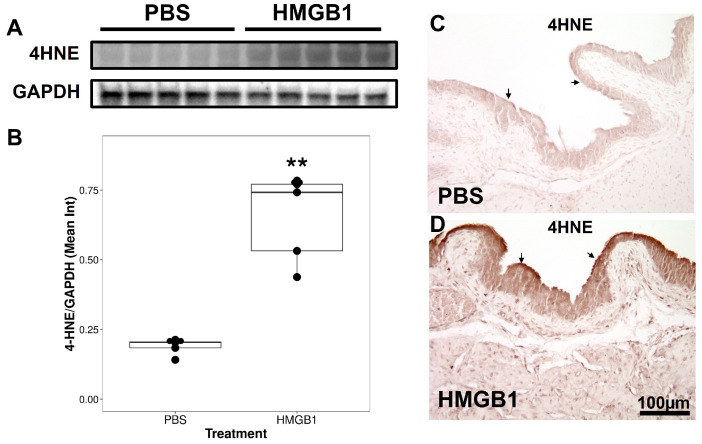
Western blotting (WB) and immunohistochemistry (IHC) analysis of 4HNE and GAPDH expression in bladder tissue. (**A**) Representative bands of Western blot showed the expression of 4HNE and GAPDH in the control group and the disulfide HMGB1 treatment group. (**B**) Statistical analysis of 4HNE staining in bladder. ** =p<0.01. Values are the ratios of 4HNE and GAPDH calculated after densitometry. Means ± SEM (n = 5). (**C**,**D**) Representative images of 4HNE IHC in control (PBS) versus disulfide HMGB1 post 1 h intravesical treatment. Scale bar: 100 μm. The arrows point to urothelium.

**Figure 4 cells-12-01440-f004:**
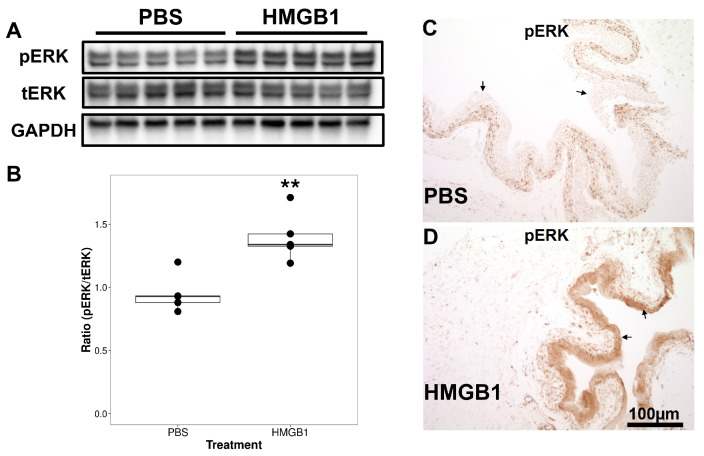
Western blotting (WB) and immunohistochemistry (IHC) analysis of phospho-ERK1/2 (pERK), total ERK1/2 (tERK) and GAPDH expression in bladder tissue. (**A**) Representative bands of Western blot showed the expression of pERK and tERK in the control group and the disulfide HMGB1 treatment group with GAPDH as a loading control. (**B**) Statistical analysis of pERK and tERK staining in bladder. ** p<0.01. Values are the ratios of pERK/tERK calculated after densitometry. Means ± SEM (n = 5). (**C**,**D**) Representative IHC images of phospho-ERK1/2 in control (PBS) versus disulfide HMGB1 post 1 h intravesical treatment. Scale bar: 100 μm. The arrows point to urothelium.

**Figure 5 cells-12-01440-f005:**
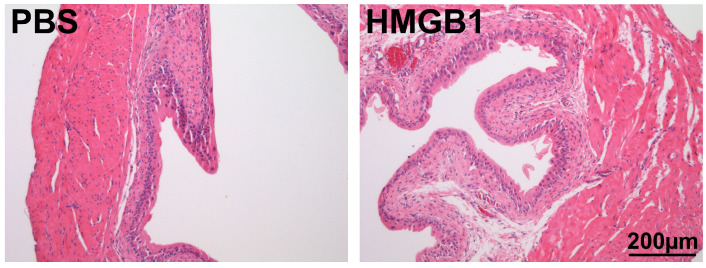
Histological changes after 1 h intravesical treatments. Bladder treated with intravesical PBS showed no signs of edema or inflammation from examination of histological sections (H&E). Some of the bladders in the group treated with HMGB1 showed mild submucosal edema (1.4 ± 0.6) and inflammation (1.1 ± 0.5). This difference was not statistically significant (*t*-test, p=0.07).

**Figure 6 cells-12-01440-f006:**
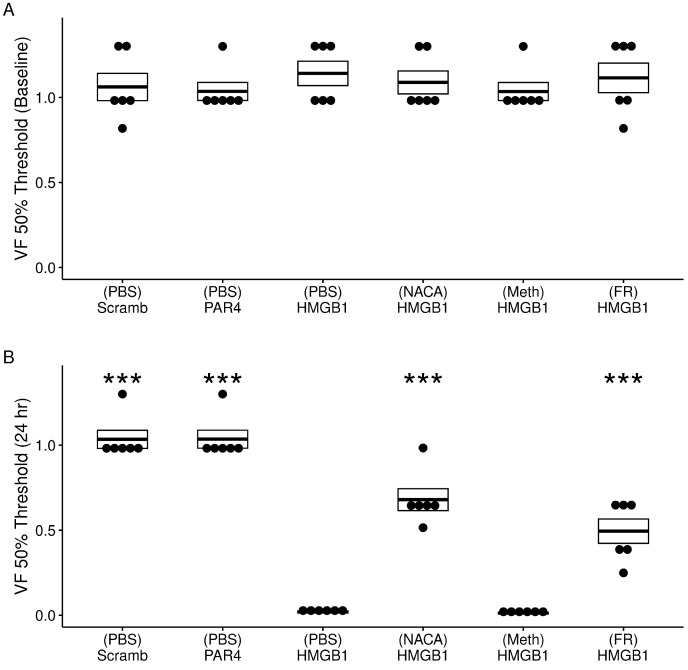
Effects of different treatments (N = 6/group) on HMGB1-induced bladder hyperalgesia (BHA) in MIF knockout mice. (**A**) No difference in 50% von Frey (VF) threshold score from lower abdominal mechanical stimulation using VF monofilaments was observed at baseline across all treatment groups (F = 0.79; ns). (**B**) Significant differences were observed in VF threshold post 24 h treatment F = 111.7, *p* = 2${}^{-16}$. Mice in the (PBS)-HMGB1 (pain group) showed a profound decrease in VF threshold score when compared to the control no pain groups: (PBS)-scrambled peptide group and (PBS)-PAR4 group. Pre-treatment with NACA (an ROS scavenger) or with FR180204 (FR, an ERK1/2 selective inhibitor) partially prevented HMGB1-induced decrease in the VF 50% threshold. Pre-treatment with methyl cellulose (Methcell) had no effect on HMGB1-induced BHA. *** *p* < 0.001; ns, not significant at *p* < 0.05.

**Figure 7 cells-12-01440-f007:**
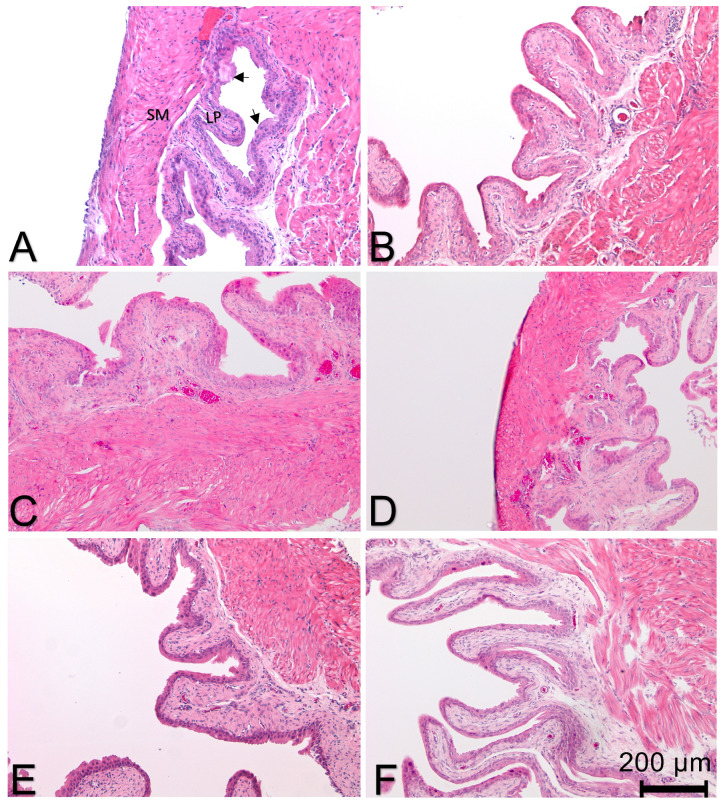
Effect of different pre-treatments (indicated in parentheses) and treatments on bladder inflammation. No significant changes in bladder edema or neutrophil infiltration were noted. (**A**) (PBS)-scramble Treatment. Arrows indicate urothelium; (**B**) (PBS)-PAR4 treatment; (**C**) (PBS)-HMGB1 treatment; (**D**) (NACA)-HMGB1 treatment; (**E**) (Methcell)-HMGB1 treatment; (**F**) (FR)-HMGB1 treatment.

**Figure 8 cells-12-01440-f008:**
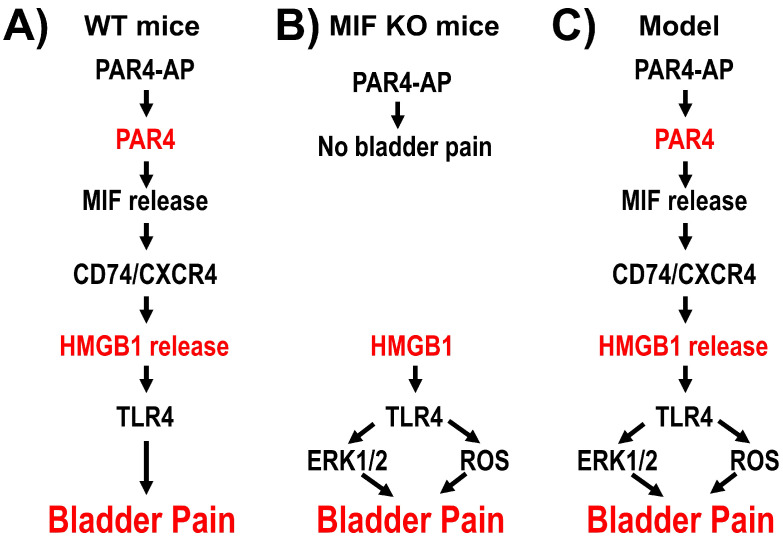
Schematic diagram showing induced bladder pain. (**A**) From our previous studies in wild type (WT) mice, PAR4-activating peptide (AP) induces bladder pain through release of urothelial MIF and HMGB1. (**B**) In MIF null mice, PAR4-AP failed to induce bladder pain, but HMGB1 induced bladder pain with the contribution of ERK1/2 activation and oxidative stress (current results). (**C**) PAR4-induced bladder pain model: upon bladder insult or PAR4 activation, urothelial MIF is released and binds to receptors CD74/CXCR4. Activated MIF receptors promote the release of urothelial HMGB1 that binds to intravesical TLR4 and leads to ERK activation and ROS accumulation, further inducing bladder pain.

**Table 1 cells-12-01440-t001:** Effect of intravesical treatments on awake micturition parameters. Mean ± SEM. N = 6 per group.

Intravesical Pretreat.	Intravesical Treat.	Volume (μL)	Frequency (Voids/3 h)
PBS	***Scramb** **peptide** **(control)***	348 ± 37.1	2.0 ± 0.5
PBS	* **PAR4** *	421 ± 32.3	1.5 ± 0.2
PBS	* **HMGB1** *	345 ± 45.5	2.0 ± 0.3
NACA	* **HMGB1** *	286 ± 45.4	2.2 ± 0.4
Methcell	* **HMGB1** *	292 ± 67.7	1.7 ± 0.3
FR	* **HMGB1** *	365 ± 49.0	1.3 ± 0.2

**Table 2 cells-12-01440-t002:** Effect of intravesical treatments on bladder edema and inflammation. Mean ± SEM. N = 6 per group.

Intravesical Pretreat.	Intravesical Treat.	Edema	Inflammation
PBS	* **PAR4** *	0.5 ± 0.1	0.3 ± 0.2
PBS	* **HMGB1** *	0.3 ± 0.2	0.5 ± 0.3
NACA	* **HMGB1** *	0.7 ± 0.3	0.9 ± 0.2
Methcell	* **HMGB1** *	0.7 ± 0.5	0.5 ± 0.2
FR	* **HMGB1** *	1.5 ± 0.3	1.3 ± 0.3

## Data Availability

Data are provided as Appendix A.

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
