# Peer review of "Urothelial Oxidative Stress and ERK Activation Mediate HMGB1-Induced Bladder Pain"

_cells, 2023, doi:10.3390/cells12101440_

Round 1

Reviewer 1 Report (Previous Reviewer 2)

The revised manuscript is a significant improvement compared to the original version. The additional data strengthen further the authors' conclusions and provide better support for their hypothesis.                                                      

Author Response

Thank you for your comments.

Reviewer 2 Report (Previous Reviewer 1)

The authors showed that HMGB1 is an independent factor in bladder pain pathology and that PAR4/MIF axis is dependent on the HMGB1 pathway to cause bladder pain in a genetically modified animal model of IC/BPS, which is indeed an important aspect in the pathophysiology of bladder pain. However, most of the findings are re-iterations of already reported studies about this pathway. In addition, the authors have frequently cited their previous work which seems to be very much similar to the current work (10.3389/fnsys.2022.882493) except for the selection of the animal model. The differences need to be highlighted as self-citation is obvious (the discussion chapter started by referring to the previous study and repeating the aim!). Unless this is made clear to the reader, I am afraid that this paper can not be considered a stand-alone scientific report. 

Author Response

Thank you for your comments.

We apologize that we did not clearly state how the current study is different from previous published ones.

We have extensively modified the manuscript, principally the Introduction and Discussion sections, to clearly state how the current study differs from previous findings. We have also changed the title of the manuscript.

We hope these changes have made our aims for the study clearer.

Thanks.

Round 2

Reviewer 2 Report (Previous Reviewer 1)

The authors have adequately modified the text in a way to show the added benefit of the submitted work as a continuation of their previous work.

I have no comments to add.

Thank you  

This manuscript is a resubmission of an earlier submission. The following is a list of the peer review reports and author responses from that submission.

Round 1

Reviewer 1 Report

The authors studied the role of HMGB1 and MIF in bladder pain which is an important condition to investigate in the field of urology. The single novel finding here is that HMGB1 can independently cause bladder pain in MIF KO mice, with reiterating the downstream role of ROS and ERK1/2 in HMGB1-caused bladder pain.The manuscript is well written and adequately presented. However, looking at the study design, results and discussion, the reader can feel directly that the paper can not stand alone without reading "the authors previous publications/studies" about the topic, and the lack of a true control (wild type) for reference.

Here are points need to be addressed by the authors:

- The study was conducted on KO rodents only, throughout the manuscripts the authors have made references to WT results from previous studies without giving detailed description about their previous findings. If the current study was done only on KO rodents, I believe a true baseline or final conclusions can not be made unless the authors made clear comparisons to their previous WT controls.

- Further details about the histological examination (namely the selection of the tested field, how this was decided, which part of the bladder was excluded from the histological examination "dome, base...").

- Regarding the VSOP test, could the authors justify the selection of 3 hours as enough to observe the micturition frequency and volume, given that most groups had done 1-2 voids. Did the authors observe any abnormal behaviour of voiding with this test, central vs peripheral voiding behaviour, which is an important variable that can be derived from this test?

- Are there baseline histological and functional differences in bladder morphology and physiology between WT and the used KO model. If available, kindly highlight that. 

Thank you

Reviewer 2 Report

The only support for the role of ERK1/2 involvement is through measuring bladder hyperalgesia and using inhibitor.

Could you do some IHC or western blots for phospho-ERK1/2?

Measuring ROS in the context of ERK1/2 inhibition?

What about the role of other stress kinases, like p39 or JNK1/2?